# OpenReview forum: "GT-HarmBench: Benchmarking AI Safety Risks Through the Lens of Game Theory"
_ICLR.cc/2026/Workshop/AFAA — AFAA 2026 Poster_

### Official Review · Reviewer_ZZjJ · 2026-02-20
**Interesting**

**Rating:** 4
**Confidence:** 3

**Summary:**

The paper presents a benchmark dataset containing game theoretic problems.
These are useful to evaluate agents in multi-agent contexts.
It analyzes how well existing LLM based agents perform on these benchmarks.
The paper also studies on how interventions (reformulating the problem) can help the systems to perform better.

**Strengths:**

* I assume that after acceptance and de-anonymization the authors will add a link to a well-documented publicly accessable benchmark dataset
* The text is well-written and understandable
* The content seems novel to me (but I'm not an expert on LLM benchmarking)
* Most results seem plausible, but I can't verify their soundness.  It would be good if the authors would provide their code to reproduce their experiments.

**Weaknesses:**

* The results are not really unexpected
* The technical elaboration seems plausible at a high level, but the text could contain more details.  Even if code and data would be available, the text would benefit from more specifications on the experimental setup to make the work fully reproducible.

---

### Official Review · Reviewer_XJKZ · 2026-02-20

**Rating:** 3
**Confidence:** 4

**Summary:**

**Summary**: This paper introduces a novel benchmark for evaluating LLM safety in multi-agent game-theoretic settings, filling a meaningful gap in existing safety evaluation frameworks. The methodology is systematic, and the mechanism design analysis offers practical value, though the restriction to symmetric 2×2 games and synthetic self-play evaluation limits real-world applicability. Overall, a solid contribution with clear directions for extension, but ecological validity concerns temper enthusiasm.

**Strengths:**

**Strengths**: The paper addresses a genuine gap by systematically evaluating LLM safety in multi-agent strategic settings, an underexplored area. The benchmark design is methodologically sound, grounding scenarios in the MIT AI Risk Repository and covering six canonical game types. The mechanism design experiments offer actionable insights, and the bias analysis (order effects, game-theoretic framing) is well-executed.

**Weaknesses:**

**Weaknesses**: The restriction to symmetric 2×2 games, while justified, significantly limits ecological validity. The self-play evaluation conflates individual reasoning quality with coordination ability. Using LLM-as-judge for reasoning classification introduces circularity concerns. The human validation study (n=30, 2 annotators) is underpowered.

**Limitations**: Scenarios are synthetically generated, and transfer to naturalistic multi-agent deployments remains undemonstrated. Mechanism designs are purely prompt-based, limiting practical applicability.

---

### Official Review · Reviewer_HDzH · 2026-02-23
**GT-HarmBench: Benchmarking AI Safety Risks Through the Lens of Game Theory**

**Rating:** 5
**Confidence:** 5

**Summary:**

This paper provides a new benchmark consisting of 2009 high stake scenarios designed to evaluate the risks of multi agent interactions. This work uses game theory to demonstrate failure modes and contradictions.
Across 15 frontier models this work found that the socially accepted outcome was achieved only 62% of the times which frequently led to catastrophic outcomes. They also propose different techniques to achieve the social benefitting scenarios by 18%.

**Strengths:**

1) Progress in the right direction of multi-agent safety. Other works such as Harmbench exist but only showcase a single agent setting, this sets a new standard by showcasing how multiagent setup can be studied.
2) By using a principled reduction from 144 2x2 games to six canonical games the authors ensure a tractable yet exhaustive analysis of all strategic tensions possible in two-player binary-choice settings.
3) The paper does more than just benchmarking: It also gives the "mechanism design" to actually increase the number of times the socially beneficial outcome is selected.
4) The other novel outcome of this study is also showcasing how framing the prompt as in game theory increases the nash equillibrium convergence.

**Weaknesses:**

1) In a real world setting, the opponents incentives are usually not known so its tough to always frame it using game theory.\
2) The study is limited to 2x2 games - a deeper study might now be required on more complex games with longer horizons based on the success of this study.

---

### Meta-Review · Area_Chair_F4DP · 2026-02-23

**Recommendation:** Main Papers Track
**Confidence:** 4

**Metareview:**

The paper proposes a benchmark based on game theory which prompted slightly mixed sentiments among reviewers, but they all agree that the work is technically sound (pending code sharing).
I think the paper fits the workshop theme and could prompt interesting conversations and also benefit from discussion with attendees on how to improve the realism and validity of the benchmark.

---

### Decision · Program_Chairs · 2026-03-02

Accept (Poster)